# The center cannot hold: A Bayesian chronology for the collapse of Tiwanaku

Erik J. Marsh[1]*, Alexei Vranich[2], Deborah Blom[3], Maria Bruno[4], Katharine Davis[5], Jonah Augustine[6], Nicole C. Couture[7], Santiago Ancapichún[8], Kelly J. Knudson[9], Danijela Popović[10], Gianni Cunietti[11]

1 CONICET, Laboratorio de Paleoecología Humana, Instituto Interdisciplinario de Ciencias Básicas, Universidad Nacional de Cuyo, Mendoza, Argentina, 2 Center of Andean Studies, University of Warsaw, Warsaw, Poland, 3 Department of Anthropology, University of Vermont, Burlington, Vermont, United States of America, 4 Department of Anthropology & Archaeology, Dickinson College, Carlisle, Pennsylvania, United States of America, 5 Department of Anthropology and Sociology, Ursinus College, Collegeville, Pennsylvania, United States of America, 6 Department of Anthropology, University of Wisconsin, Madison, Wisconsin, United States of America, 7 Department of Anthropology, McGill University, Montreal, Quebec, Canada, 8 Centro de Investigación GAIA Antártica, Universidad de Magallanes, Punta Arenas, Chile, 9 Center for Bioarchaeological Research, School of Human Evolution and Social Change, Arizona State University, Tempe, Arizona, United States of America, 10 Centre of New Technologies, University of Warsaw, Warsaw, Poland, 11 Facultad de Filosofía y Letras, Universidad Nacional de Cuyo, Mendoza, Argentina

* emarsh@mendoza-conicet.gob.ar

**Data Availability Statement:** All relevant data for the radiocarbon dating and modeling are within the manuscript and its Supporting Information files.

## Abstract

The timing of Tiwanaku's collapse remains contested. Here we present a generational-scale chronology of Tiwanaku using Bayesian models of 102 radiocarbon dates, including 45 unpublished dates. This chronology tracks four community practices: residing short- vs. long-term, constructing monuments, discarding decorated ceramics, and leaving human burials. Tiwanaku was founded around ~AD 180 and around ~AD 600, it became the region's principal destination for migrants. It grew into one of the Andes' first cities and became famous for its decorated ceramics, carved monoliths, and large monuments. Our Bayesian models show that monument building ended ~AD 720 (the median of the ending boundary). Around ~AD 910, burials in tombs ceased as violent deaths began, which we document for the first time in this paper. Ritualized murders are limited to the century leading up to ~AD 1020. Our clearest proxy for social networks breaking down is a precise estimate for the end of permanent residence, ~AD 1010 (970–1050, 95%). This major inflection point was followed by visitors who used the same ceramics until ~AD 1040. Temporary camps lasted until roughly ~AD 1050. These four events suggest a rapid, city-wide collapse at ~AD 1010–1050, lasting just ~20 years (0–70 years, 95%). These results suggest a cascading breakdown of community practices and social networks that were physically anchored at Tiwanaku, though visitors continued to leave informal burials for centuries. This generation-scale chronology suggests that collapse 1) took place well before reduced precipitation, hence this was not a drought-induced societal change and 2) a few resilient communities sustained some traditions at other sites, hence the chronology for the site of Tiwanaku cannot be transposed to all sites with similar material culture.

The orthomosaic of Tiwanaku is available here: https://osf.io/v6j7n/.

**Funding:** Excavation and radiocarbon dating was funded by the following institutions. To Alexei Vranich: National Science Foundation (IIS-0431070, BCS-0415914) To Deborah Blom: National Science Foundation (BSC-1317184), The University of Vermont's College of Arts and Science, The Wenner Gren Foundation for Anthropological Research To Nicole C. Couture: Social Sciences and Humanities Research Council, Standard Research Grant (410-2006-1806), The Canadian Foundation for Innovation Infrastructure Grant (202493), Fond Québécois de Recherches sur la Société et la Culture, Nouveau Chercheur Grant (116296), McGill University Faculty of Arts Research Fund To Kelly J. Knudson: National Science Foundation (BCS-1523209) To Danijela Popović: National Science Centre, Poland. 2014/15/D/NZ8/00285.

**Competing interests:** The authors have declared that no competing interests exist.

## Introduction

'Collapse' evokes romanticized visions of the end of civilization, which have long been part of popular narratives. Collapse is usually the last of five stages, which Thomas Cole illustrated vividly in a series of five paintings from the 1830s, *The Course of Empire.* Each painting shows a stage in a Rome-inspired city, ending with overgrown ruins. In the nineteenth century, early scholars transposed this template to Tiwanaku, high in the Andes of South America. Once called the "American stonehenge" [1:277], the site is littered with carved blocks and eroded monuments on a "bleak, windswept, bare, and frigid land" [2:272]. For generations, this juxtaposition has stoked the mystery of how such a city could have supported a large population or been a "seat of dominion" [1:299–300]. Early speculations and the first archaeological accounts relied heavily on idealized narratives, which retained the five-stage narrative to tell Tiwanaku's story [3, 4]. The final stage usually featured an external force, such as foreign invasion or climatic disaster, that brought a dramatic and sudden end to a once-glorious civilization.

These eye-catching tales are prevalent in popular writing (most prominently [5]), but they tend to "dramatize long-past events [and] push human actors into the background" [6:1]. Archaeological research suggests that most of these accounts overlook continuity and resilience at human time scales [7–15]. Simplified narratives of collapse run into three major problems [16:134]. First, they assume societies end decisively, all at once, conflating "profound societal change with . . . biological extinction" [17]. In well-documented collapses, some parts of society are drastically restructured while others remain resilient [7]. The 'collapse' label may not appear until generations later, when commentators can take significant liberties to tell a more compelling story [6, 18]. Second, these narratives often lean on external factors while underplaying internal sources of strife, not to mention the human capacity to adjust to new climate and social configurations [11, 19]. Finally, and most relevant for this paper, imprecise chronologies leave ample room for apocalyptic scenarios to overshadow data [8, 20–22].

Similar ideas are echoed in previous approaches to Tiwanaku's collapse. Tiwanaku is usually treated as the capital of a politically-unified state that stopped all at once, despite imprecise chronologies [23]. Chronology is closely tied to decorated ceramics, but this simplification runs the risk of overlooking the human activities that created the material record. Many interpretations prioritize external factors, for example, an invasion by the present inhabitants of the area, the Aymara, was a popular explanation in the 1980s but has since been discarded [24, 25:177]. More prominently, the role of drought has been hotly contested for decades [see review in 26]. Lake core data identify a drought, which been proposed as the culprit of the collapse of both the city and state of Tiwanaku [23, 27, 28]. While climate chronologies remain imprecise, they suggest the drought began in the twelfth century, over a century after archaeologists suggest the site was abandoned [29–31]. Besides external factors, internal social tensions certainly had a role to play [25:200]. Some dramatic scenarios describe elites sacrificing captives, in addition to razing elite residences and ritually 'killing' monolith beings [25:192, 32:445, 33:303].

In order to develop a better picture of human responses to external and internal factors, we first need a generational-scale chronology, which is the goal of this paper. This is crucial because blurry sequences of events leave room for narratives to rely more on model expectations than data [20, 22, 34]. A generational-scale chronology is the crux for tracking cause and effect relationships and human responses to external factors. It also allows us to assess the rate of change and distinguish between two differing views, that collapse was "explosive" [29] or alternatively, "a long time of troubles" [25:188]. To do this, we need to clarify what we mean by collapse [14, 16, 18, 26, 35]. We might look to the end of elite burials and residences [36:367–369], but this overlooks the rest of the community. Others have defined collapse as a

rapid loss of sociopolitical complexity for an entire polity [11, 14:4, 26:84–88], but this does not have a direct material correlate and assumes synchronized changes at multiple sites. Instead, we take a site-by-site approach. Our argument for collapse applies only to Tiwanaku, not all sites with similar ceramics. At Tiwanaku, we identify discontinuities in long-term community practices [14:4, 26:87]. These practices can be documented by material evidence for what people were doing and for how long. Instead of the value-laden question "what went wrong?", we seek answers from a more useful question: "which practices changed quickly?" [6:7–8, 18]. To do this, we reassess excavation data to independently track four key community practices at Tiwanaku: residing short- vs. long-term, constructing monuments, making and discarding decorated ceramics, and depositing human bodies.

First, we distinguish between short-and long-term occupations (Fig 1), since the abandonment of an urban center is a simple yet crucial indicator of the end of sustained face-to-face interactions [35]. This would also mean the end to nearly all local political activity and communication with other sites, since there were no written documents in the prehispanic Andes. Long-term residents at Tiwanaku left dense and diverse middens with undecorated pottery for cooking and storage, decorated sherds for serving food, worked bone tools, animal bones, and

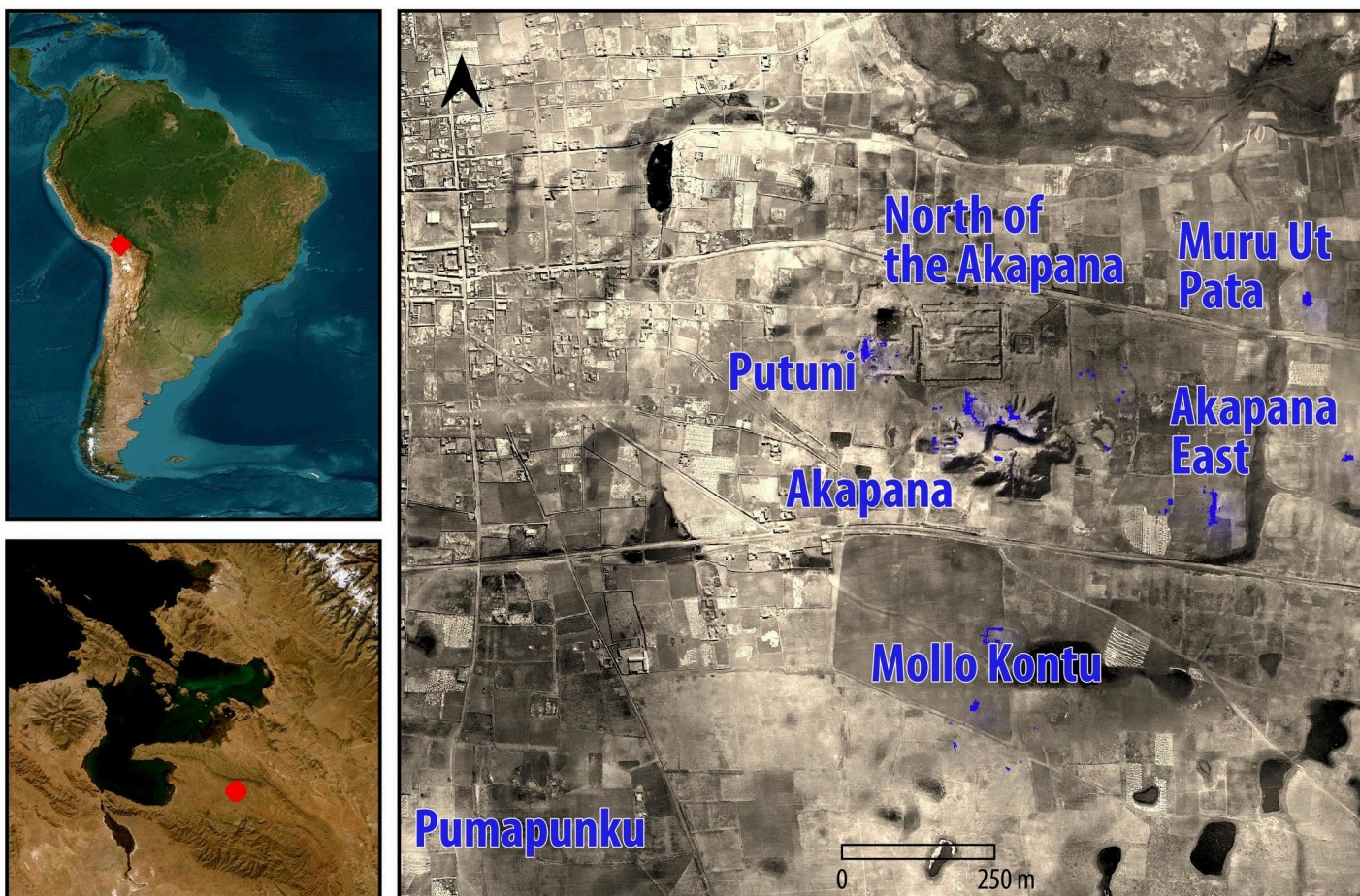

**Fig 1. Tiwanaku's location.** Shown as a red dot in South America (a) and in the southern Lake Titicaca Basin (b, basemaps are from OpenStreetMap sources accessed with the tool MapTiler in QGIS 3). Labels identify the main excavation sectors mapped in blue (S2 File). The base map is an orthomosaic of aerial photos from the 1930s (S5 File).

plant remains (see details in [37–39]). They placed offerings under wall foundations and built residential compounds with central patios that followed the same alignment throughout the site. In contrast, temporary visitors tended to leave isolated hearths, reuse existing materials, cut into earlier walls and deposits, and forgo building structures with stone foundations that could support adobe superstructures. Temporary visitors probably left much of the refuse at Tiwanaku [39], contrary to general expectations for cities and especially capitals which tend to assume that most remains were left by permanent residents.

Second, we refine the chronologies of two of the largest monuments at the site, the Akapana and the Pumapunku (Fig 2). The Akapana is the site's largest moment, a 17-m tall stepped platform with seven terraces, some with interior retaining walls and stone facades. The Pumapunku is a smaller stepped platform renowned for oversized stone blocks and precisely cut ashlars. To refine the chronologies of these monuments, we distinguish dated contexts related to their construction from offerings and burials left on or near them.

Third, we compile all dated contexts that include Tiwanaku's iconic redwares (Fig 3). These ceramics are among the region's best-made vessels and feature elaborate iconography [42].

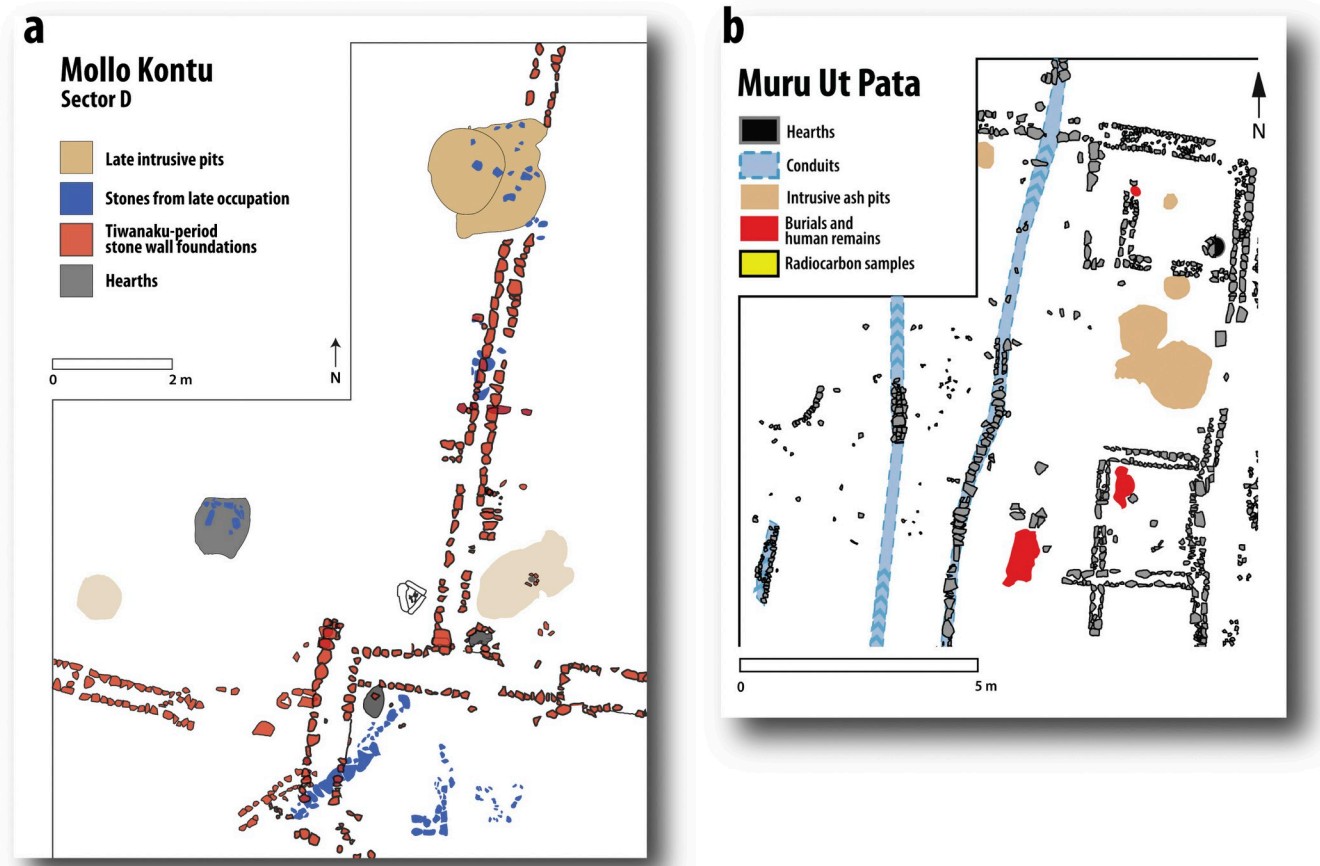

**Fig 2. Examples of residential occupations.** a: plan drawing of Mollo Kontu. b: Plan drawing of Muru Ut Pata (see S2 File). For photographs of Akapana East, structure 5, see [42:Figs 10.10 and 10.19,43]. For photographs of Akapana East 1M, level 3, see [25:Fig 10.6a,43], a late structure with redwares, interpreted here as a temporary occupation that reused existing blocks. The double-coursed stone foundations aligned a few degrees west of north follow the site-wide pattern for Tiwanaku-period residential architecture. In most cases, walls enclose smaller structures that face an open area, forming a residential compound or patio group, the standard spatial layout at the site [41]. These stone foundations helped slow rising damp from degrading adobe walls. Later, alignment changed with improvised structures, for example, in Mollo Kontu, as in the stones in blue in (a) and the reused single-file blocks in Akapana East 1M. It is unclear what superstructures these buildings had, if any.

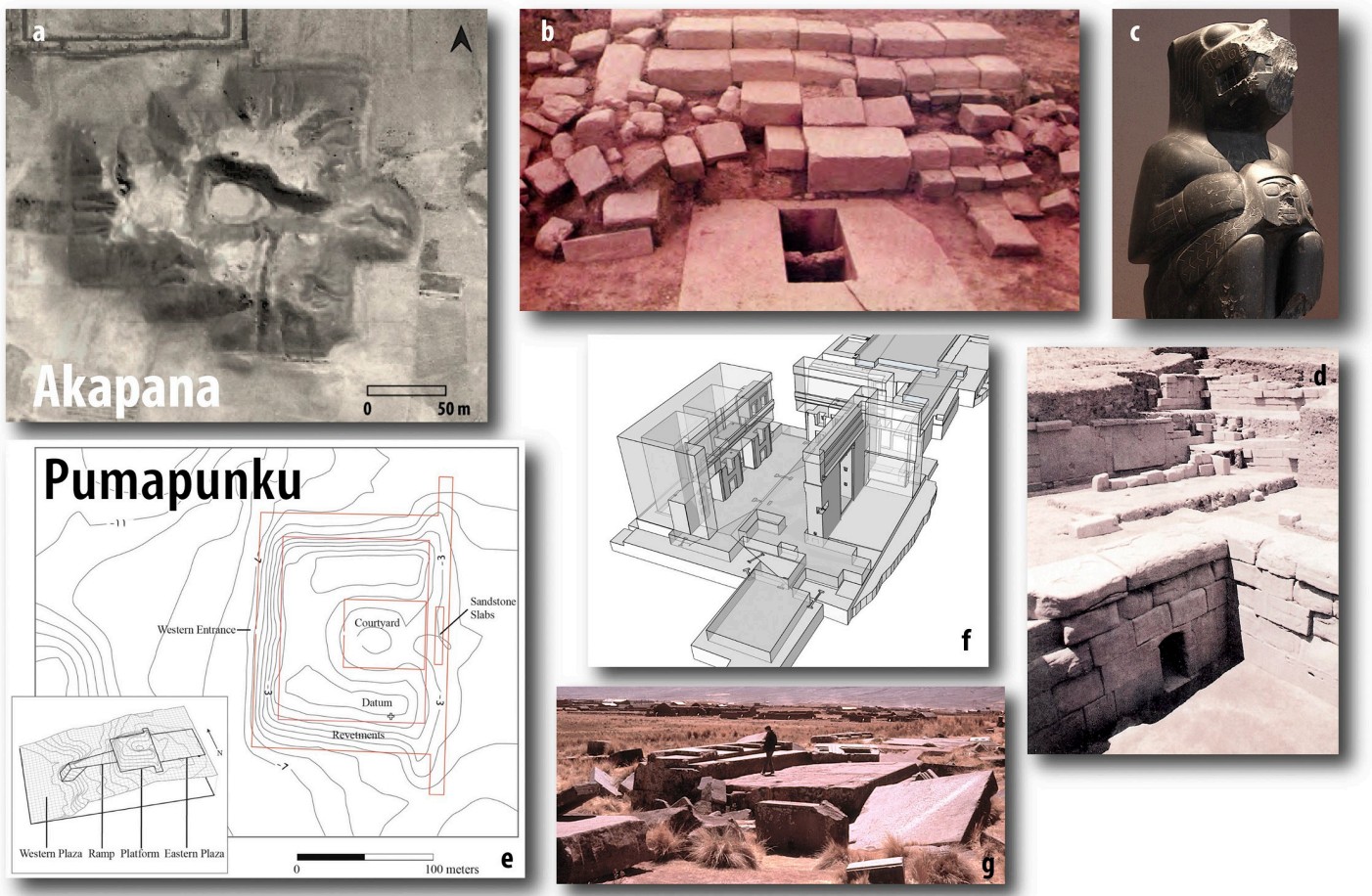

**Fig 3. Principal Tiwanaku-period monuments: the Akapana (a–d) and Pumapunku (e–g).** a: the Akapana, visible on the orthomosaic of aerial photos from the 1930s (see S5 File). Dimensions: 197×203×16.5 m [44:22]. It just south of the earlier Kalasasaya and Sunken Temple (summarized in 45:106–109; see S2 File). b: the stepped entrance on the western side of the Akapana. c: a statue that once stood next to the entrance, called the Chachapuma, holding a trophy head [46]. d: area of the monument with the mouth of conduit for internal drainage system [44:Photo D, 47]. e: outline of the Pumapunku [48:Fig 3]. f: reconstruction of the northern portion of the Pumapunku's andesite building [48:Fig 20]. g: large scattered andesite blocks that were once part of the monument. For more photographs and drawings, see [23, 44, 45, 47–52], though we should be wary of ideal reconstructions based on right angles, symmetry, and proportions [44:Fig 4, 51]. Many reconstructions follow idealized visions that tend to overlook details that do not conform to modern architectural conventions [52]. Figs 5b and 5d are republished from [44] under a CC BY license, with permission from the Instituto de Investigaciones Antropológicas, UNAM, Mexico, original copyright 1992, courtesy of Linda Manzanilla. The photographs in 5c and 5g were taken by Alexei Vranich. Fig 5e and 5f are republished from [48] under a CC BY license, with permission from Alexei Vranich, original copyright 2018.

Some of these ceramics were used to serve corn beer, a key interaction in the community events that built Tiwanaku.

Fourth, we address the changing practice of the treatment of human remains (Fig 4). We distinguish 1) formal tombs, often placed in residential areas with grave goods, 2) bodies left in refuse pits, often young individuals who died violently, and 3) informal burials without mortuary architecture or associated artifacts. Some burials follow a repeated material pattern, with the co-occurrence of human bone, animal bone, and decorated Tiwanaku ceramics.

## Materials and methods

Up to now, Tiwanaku's collapse chronology has been mostly based on informal treatments of individual radiocarbon dates with large error ranges. A telling example is a date from an elaborate offering on the Akapana (SMU-2473), interpreted as a closing ceremony during the

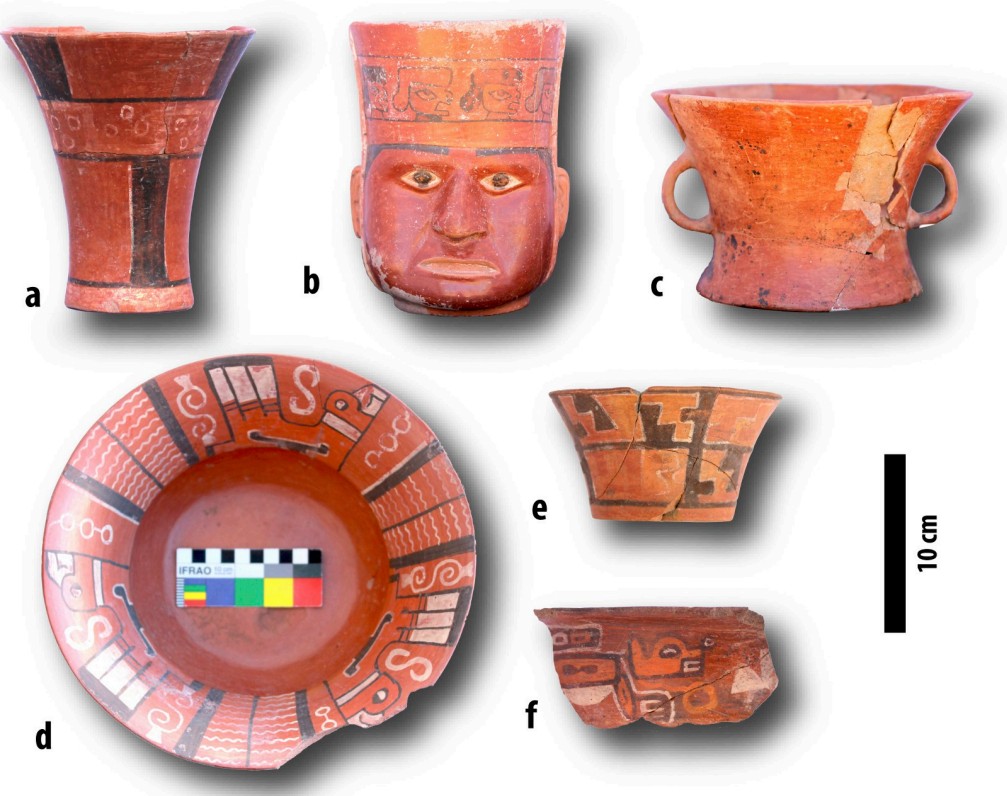

**Fig 4. Representative examples of Tiwanaku redwares.** Ancient potters fired these ceramics at high temperatures, decorated them with red slips and painted designs, and innovated a series of new forms. a: hyperboloid drinking goblet (*kero*; event A111, midden). b: portrait vessel (*wako retrato*; A99, an offering from burial A96; see Fig 7). c: pedestalled burner (*sahumador*) from the same burial as (b). d: shallow bowl (height: 11 cm), with an everted rim (*escudilla*), plan view. e: hyperboloid bowl (*tazón*; D40, ash pit). f: bowl with everted rim (*escudilla*; D36, hearth). Scale is approximate, due to minor photographic distortions. All materials are from the Mollo Kontu sector [53] except (d), which is from a mass burial north of the Akapana [54]. Photographs by J. Augustine.

twelfth-century drought [23:121, 44:253]. However, this claim is tenuous: the date's error range of ±243 years makes it uninformative. A second radiocarbon date (INAH-972) from the same context, often overlooked, is three centuries older [43:107]. We took three steps to improve our materials and methods for dealing with radiocarbon dates.

Our first step was to build a database of 102 dates for the sites, comprising 12 dates from recent genetics studies, 45 unpublished dates, and 45 published dates (S1 Table). We include 28 dates run at two Polish laboratories (Wk and Poz); 22 of these are duplicates run on the same individual, which increases precision. We present 26 unpublished dates from Mollo Kontu, which was excavated by the project Jach'a Marka led by Couture, Blom, and Bruno [52:163, 169, 54:70, 58:Table 3.1]. Some of these were selected and submitted by Knudson, Blom, and Janusek, who also ran unpublished dates from three other sectors. For each dated context, we assessed associated material culture and depositional sequences using unpublished reports, field forms, and photographs from multiple projects (S1 Table, S2 File).

Second, we used a site-specific calibration curve mixture specific to Tiwanaku, which reflects the fact that air parcels reach the site from both the Southern and Northern Hemispheres [59]. The relative proportions are based on a HYSPLIT climate model [60–62]. We implemented this in OxCal (S1 File) as a bespoke mixture of SHCal20 and IntCal20 [63, 64].

Third, we leveraged this new calibration curve and contextual details to build Bayesian models in OxCal 4.4 [65] (S1–S4 Files). Bayesian models allow us to scale up from individual dates, that is, to see the forest despite the trees. These models can identify trends in large sets of radiocarbon dates, and for this reason, have driven the third radiocarbon revolution [66]. Here we bring this revolution to Tiwanaku, updating and expanding on previous models [29, 67, 68]. Bayesian models can significantly reduce error ranges, especially when there are multiple dates from the same context or clear stratigraphic superpositions. Modeled dates are reported in italics and medians with a leading tilde; this simplification improves readability but full error ranges remain important for making interpretations (S1 Table).

All necessary permits were obtained for the described study, which complied with all relevant regulations. Permits to excavate and export radiocarbon samples were obtained from the Bolivian government (la Unidad Nacional de Arqueología, la Dirección Nacional de Arqueología, La Unidad de Arqueología y Museos) under permits 086/07, 023/08, 052/2016, and 086/2016. Additional information regarding the ethical, cultural, and scientific considerations specific to inclusivity in global research is included in the S6 File.

## Results

The key results are the boundaries of each Bayesian model (S2 File). These estimates tell us, with error ranges, when specific practices started and stopped in each of the site's main sectors: Akapana East, Muru Ut Pata, Mollo Kontu, the Putuni, the Pumapunku, and the Akapana (Fig 5, S2 File). We cross-referenced these sector models into site-wide composite models that provide estimates of when specific human activities started and ended: 1) permanent and temporary residential occupations where people slept, ate, and socialized, 2) monuments that residents and visitors built through community-wide collaboration, 3) decorated ceramics people made and discarded for eating, drinking, marking status, and communicating through iconography, and 4) burying ancestors in formal tombs with grave goods and for others, carrying out much more expedient depositions.

### Akapana East

Extensive horizontal excavation produced one of the best documented residential areas [37, 38, 42, 40, 69]. We grouped 15 dates into three phases. The first phase comprises three dates, which have modeled medians of ~AD 620–660. Most of the material is from the second phase, which included extensive kitchen refuse, residential architecture, and decorated Tiwanaku ceramics. Six similar dates suggest this phase spanned ~AD 800–970 (First and Last queries). Next, the end of long-term occupation is defined by charred roof thatch found on living floors. In the Bayesian model, this boundary is ~AD 1020 (910–1140, 95%). The adobe walls had fallen by the time visitors made temporary camps here [25:189, 32:441]. This final phase of occupation is defined by three dates, with medians of ~AD 1100–1160. This suggests that some time passed between the final occupation of the building and the ephemeral occupation [25:189, 32:441]. Visitors who stayed here cut pits into previous deposits, making date–artifact associations less clear.

### Muru Ut Pata

This residential area is similar to Akapana East. These unpublished excavations, directed by K. Davis, uncovered a contiguous area of 327 m². As in other residential sectors, a series of cell-like rooms surround an open patio. The area is dense with middens, animal bone, decorated and undecorated ceramics, and 17 burials. There are three precise radiocarbon dates [70]. The first two both model to ~AD 920. The samples are from the main occupation and living floors

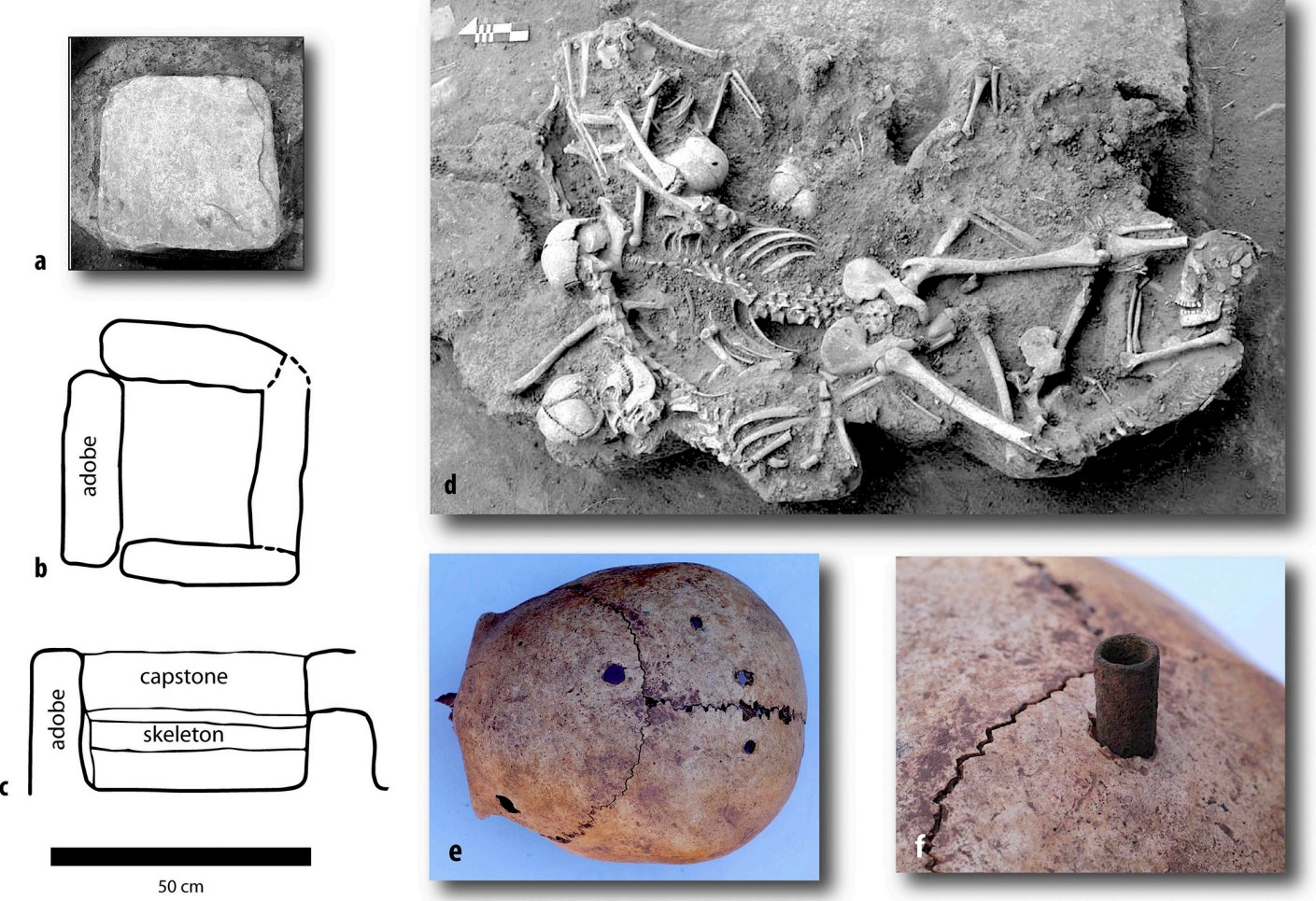

**Fig 5. Tombs and violent deaths.** a, b: The capstone for a rectilinear tomb with adobe walls, plan view, from Mollo Kontu (events A96 and A97, Beta-275871, individual MK-05144 [55]:Fig 4.18–19], c: profile view of a tomb from Mollo Kontu (events A102/A103, Beta-275872, individual MK-05404) [55]:Fig 23]). These small tombs were made for infants (individuals MK-05144 and MK-05404). This tomb architecture is diagnostic of the Tiwanaku period [56]:183–185]. Scale bar only applies to a–c. d: Photograph of a mass burial north of the Akapana, taken while exposing the bones of individual 12, center, found face down with his arms and legs spread out. This young adult male had a stature of 171.5 cm and unhealed cranial wounds and jaw and tooth fractures [57]:177–178, Fig 13.4]. e, f: cranium of individual 7, from the same context as (d), with four puncture holes and the likely element used, a bronze point (45×9 mm), perhaps the tip of a staff found in the same context [54,57]:173, Fig 13.15]. See [58]:21–22] for images of Individuals 3 and 4 from the base of the Headless Monolith, found face down in a trash pit. See [32]:Fig 18.5] for detail of cut marks on a right distal femur found on the Akapana. Photographs 5d–f by John Verano, used with permission.

and a shaft tomb. The third is from an ash layer that overlays the main occupation, marking the end of residential occupation ~*AD 950*. This suggests an occupation of roughly two generations. Around this time or shortly after, people buried three younger individuals over previous living surfaces with no burial architecture; two are flexed adolescents facing each other. There is a hearth by each of their heads, and they were both buried with damaged but carefully salvaged black incised ware and a polychrome kero. The third is a child left with little ceremony, face-down and limbs askew.

## Mollo Kontu

To the south of the site's core, there are two excavation areas: Mollo Kontu mound and Mollo Kontu South (Fig 5 and S3.5 in S2 File). The Mollo Kontu mound has stepped revetments

reminiscent of the larger Akapana. Many burials were placed around the mound, but there are no clear stratigraphic relationships between them to build Bayesian models. There are eight dates from seven individuals. One male was found face down with his hands behind his back and dates to ~AD 990 (890–1030, 95%; AA-107602; Feature 19) [71:221, Fig 8.37]. He and three other individuals were buried around the same time; the four medians span ~AD 940–1000. The other three dated individuals died later at ~AD 1090, 1250, and 1300 respectively.

For Mollo Kontu South, we included dates from three areas, sectors A, C, and D (S2 File). Sector A was initially thought to be a burial area, but excavators now consider this to be a badly disturbed residential area where burials were especially common. Well-defined surfaces separating the deposition of the burials allowed us to build a stratigraphic model. There were two burials with formal tombs containing decorated ceramics *~AD 680–790*. One child burial was later, *~AD 970* and a post-abandonment midden was dated to *~AD 1040*. In sector C, three individuals were found buried with grave goods, including a gold lamina, obsidian projectile points, and a beaded necklace. These are some of the site's earliest burials, with medians of ~AD 570–590.

Nearby, in sector D, excavators documented three stratigraphically distinct domestic occupations, modeled as a sequence of three phases, echoing Akapana East. The earliest occupation was only partially excavated; two dates have medians of *~AD 680* and *~AD 750*. The second occupation was notable for its well-preserved wall foundations [72]. From this stratum, five dates have clustered medians in the range *~AD 920–1000*. The model boundary suggests permanent residence in Mollo Kontu South ended *~AD 1020 (970–1110, 95%)*. The final occupation was characterized by a notably different use of space [58:85–88]. There were many intrusive ash-filled pits but scant evidence for the orthogonal compound walls that characterized earlier, permanent settlements. A late hearth from this phase was dated to *~AD 1030*. Notably, both this hearth and the post-abandonment midden in Sector A contained decorated, Tiwanaku style ceramics. These two contexts are among the latest from which redwares were recovered. Most likely, visitors repurposed, unintentionally mixed, and/or brought older ceramics from elsewhere.

## The Putuni

The Putuni monument is a courtyard defined by a low wall of reused stone located between the Kalasasaya and the Kherikala [33, 73]. The monument represents the last occupation of this important area. There are eight dates from three clear occupations, beginning with an occupation surface overlaying a clay platform. The second occupation is defined by a dense layer of refuse in a large kitchen area, a system of stone-lined drainage conduits, and an attached mortuary sector. One sample of human bone is from a female with lowland genetics [74]. Other human remains were found in an attached mortuary complex: one individual found inside a massive urn, over 1 m in diameter, in addition to a woman buried with elaborate grave goods: a beaded necklace, copper bracelets, a mirror, and a gold plate with a human face. Three dates from this occupation are very similar: *~AD 710–730*; one is slightly later, *~AD 820*.

The entire area to the west of the Kalasasaya was later razed and covered with deep layers of fill to support a 50 × 70 m courtyard. The stones that formed the courtyard wall were reused and modified; the joints only meet precisely at the face [75]. Niches are set along the interior side of the courtyard and a decapitated monolith was found near the center [45]. The adjacent patio group has the same layout as other residential areas but is more elaborate: the patio is lined with reused ashlars and paint chips, suggesting the walls were once painted in "brilliant hues of red, yellow, orange, green, and blue" [70:251]. Artifacts include decorated and

undecorated pottery, projectile points, and large jars for making corn beer, which are rare at the site. An elaborate tomb includes a gold pendant and many decorated vessels. Three dates from the tomb combine and model to ~AD 910. Stratigraphic relationships show this tomb's dates are associated with the rest of the occupation.

The end of the patio group's occupation is defined by a layer of burned roof thatching. A burned beam was dated to ~AD 950, but we do not know how much time passed between the cutting of the beam, which is the event estimated by the radiocarbon date, and the roof's collapse. This uncertainty is reflected in the wide error range for the ending boundary of the Bayesian model: ~AD 1000 (800–1220, 95%). The end of occupation may have been abrupt or even violent, based on the presence of large smashed vessels, unprocessed animal carcasses, and the burned roof thatch [25, 70:262]. Unlike other sectors, there are no signs of subsequent temporary reoccupation.

### Pumapunku

The Pumapunku is a stepped platform that stands out for its large and finely carved blocks [49, 75, 76]. Construction began ~AD 580, according to the model's starting boundary. Five dates from above and below a green and red surface provide a fairly precise construction sequence. The model interpolates the green surface at ~AD 630, a significant date because it marks the final use of quarried, carved stones. After this, masonry employs minimally-modified reused stones, both here and at the Akapana. Hence perhaps only one or two generations of masons were responsible for most of Tiwanaku's iconic stonework that featured flat planes, geometric corners, and precision fitting [49, 51]. Construction ended ~AD 710, around the same time as an informal burial, a complete human skeleton with lowland genetics (individual TW059) [74:7]. Later, people returned to leave offerings below the exterior revetments ~AD 740–770, based on three dates [75, 76]. There are two later dated events, a looter's pit ~AD 1020 and a child burial ~AD 1090 (TW1000).

### The Akapana

The dated contexts from the Akapana are predominantly burials and offerings, which were deposited on the monument's flat summit, on its terraces, and surrounding its base. Most contexts include decorated ceramics, animal bone, and human bone. There are few clear relationships to inform Bayesian models.

**Construction and early offerings.** Four dated offerings allow us to track when certain parts of the monument were built. Grouped as a single phase, the modeled medians are very similar ~AD 640–660, but error ranges are wide. There is a dated offering that was dug into the Akapana's clay foundation and another from between the retaining wall and facade of the second terrace [43, 77, 78]. The last date is from a burial on the summit near a stone conduit. This burial was placed after the Akapana had reached its final height, a complex edifice with internal drainage conduits.

**Summit occupation and offering.** There are three dated contexts from the structures on the summit of the Akapana. A small offering near a structure entrance was dated ~AD 820. Rectilinear structure foundations surround an open patio, similar to other residential areas but this also recalls the U-shaped layout of community buildings like the Kalasasaya. Contexts associated with this structure include undecorated ceramics, suggesting some residential activity, though this area was not strictly domestic [23, 43, 77]. The rooms are arranged around a patio that includes six aligned burials. The most elaborate includes a puma-head incense burner, dated to ~AD 790 [43]. Sometime after the structure was abandoned and the adobe walls melted or knocked over, an arranged offering was placed that included 14 camelid crania

and other bones, bone and copper jewelry, and seeds from local and tropical plants. Two imprecise dates from this context, mentioned above, combine to ~*AD 1000 (680–1230, 95%; SMU-2473 & INAH-972)*.

**Exterior revetments.** Four contexts were excavated and dated in the 1990s. They have wide error ranges but similar medians: ~*AD 890–980*. The first is a concentration of human and camelid bones and ceramics near a wide staircase on the west side of the Akapana [43]. The second, along the northwest side of the platform, comprises a carnivore placed at the mouth of a stone drainage conduit (Fig 2D). The last two dated contexts are on the second terrace. The lower context has polychrome ceramics, camelid bones, and the remains of two adolescent males with cut marks on their bones. One of the adolescents was found face down [32, 43]. Above this, a second context is the skeleton of a young individual laid out over the earlier ceramic smash [32, 43, 77, 78]. These bones have marks from cutting, gnawing, and sun and wind damage, indicating they were exposed for a time, and then later were covered by wind-blown deposits.

There is another set of later dates from eight individuals (four with duplicate dates). There are no clear stratigraphic relationships between dated samples but the dates are very similar. At the base of the platform, one individual was placed at the mouth of a drainage canal, echoing the above-mentioned carnivore offering. Other dated individuals are from contexts with animal bone and decorated ceramics; one is part of a context with 14 human crania. Four dated individuals at the base have medians of ~*AD 940–970* (TW001, TW006, TW060, TW065); one is later, ~*AD 1090* (TW0061), but is from an apparently disturbed context, making it less reliable. On the terraces in the same part of the Akapana, excavations recovered human remains of three individuals but no grave goods. These have medians of ~*AD 1010*, *1070*, and *1100* (I0977, TW102, I0978).

**North of the Akapana.** In an open area north of the Akapana, excavations revealed a closed stone conduit and pebble surface [56]. Embedded in the pebble surface was the skeleton of a female who died ~*AD 980* (TW097). A nearby trench also has the same pebbled surface. On this surface, there is a dense deposit of bones from two camelids and 16 humans. The humans are nearly all young, including children, and three have evidence for violent deaths, including a cranium with puncture wounds. This context is dated by two dates from one individual, which model and combine to ~*AD 1020* (TW090). Associated goods include redwares, a rare bottle in the form of a trilobite, lithic projectile points, and metal points that were probably used to make the puncture wounds (Fig 3D) [53:68].

Slightly north of the Kalasasaya, at the base of a monolith, salvage excavations revealed a dense midden with human and animal bone, undecorated and decorated sherds. There were many disarticulated human bones but no formal pits or tombs. The position of the bones suggests they were placed in the midden face down with their hands behind their back. One had Amazonian ancestry (TW056, Individual 3; 73:7). The medians of multiple high-precision dates place two human bone samples at ~*AD 930* and *940*. Overall, this context fits the timing and pattern of offerings around the base of the Akapana.

## Permanent residence

Permanent residence is our most reliable proxy for collapse. In independent Bayesian models of four sectors, Akapana East, Muru Ut Pata, Mollo Kuntu, and the Putuni, permanent occupation ended at around the same time: ~*AD 1020*, *970*, *1020*, and *1000*, respectively (Fig 6). However, these models are based on small samples of dates so error ranges are wide. We grouped these dates into a composite site-wide model, which incorporates depositional sequences from individual sectors. More generally, the model follows the site-wide

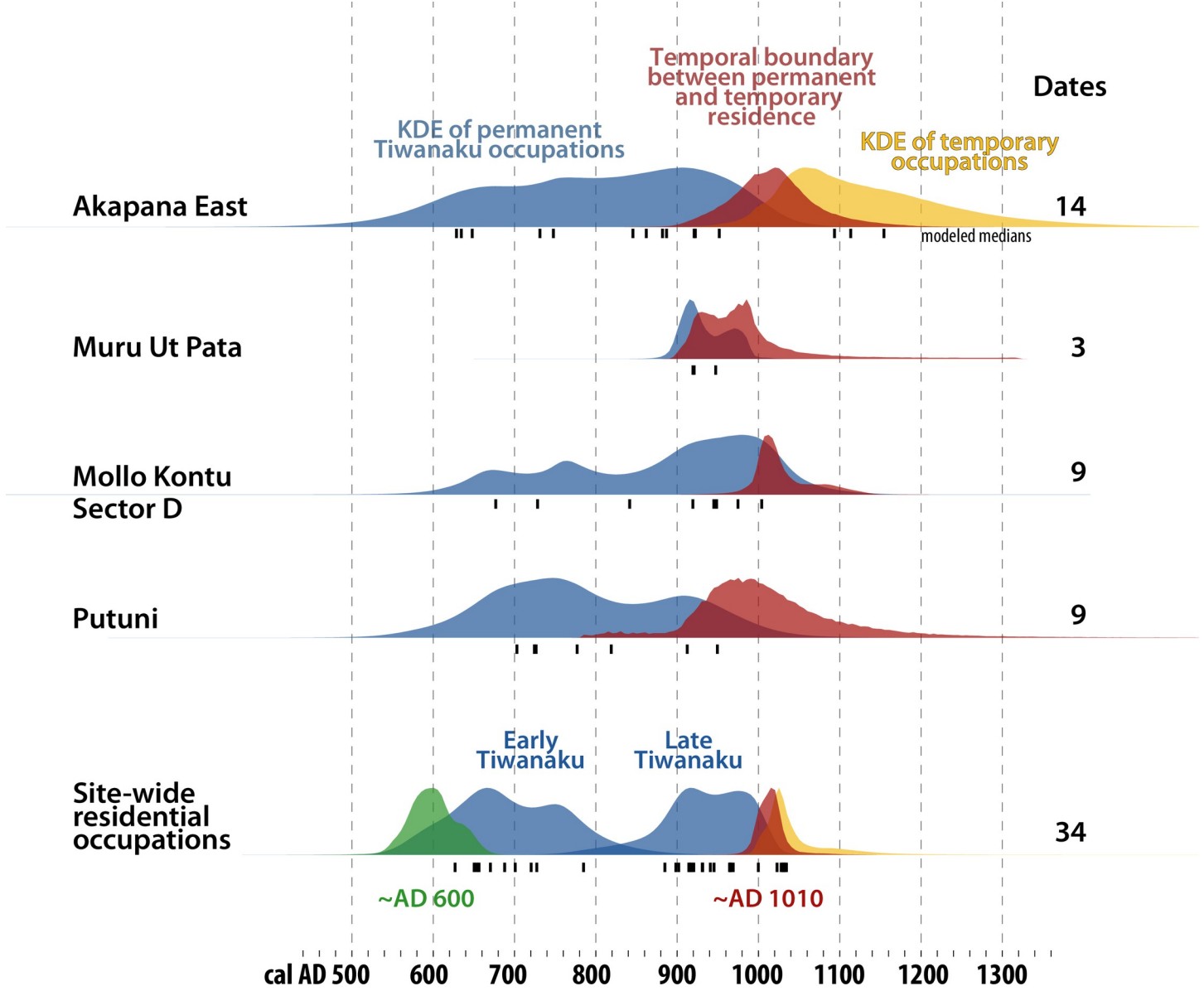

**Fig 6. Bayesian models of four residential sectors and a composite, site-wide model for permanent and temporary occupation.** The green and red curves are phase boundaries. The blue curves are KDE plots of dates associated with permanent occupation, which are robust of summarizing groups of dates [66]. The yellow curves are KDE plots of dates associated with temporary occupation. Below each KDE plot, hash marks indicate modeled medians of individual dates and contexts; the sample size for each model is listed on the right. For phases with few dated events, boundaries are not shown because they are not meaningful with such small samples. In these cases, the KDE plots can only be treated as visual guides to the distribution of sparse data.

stratigraphic pattern: Early Tiwanaku occupations lie below Late Tiwanaku occupations. Before beginning Late Tiwanaku occupations, residents razed earlier occupations, perhaps because dilapidated adobe buildings are easier to rebuild than rehabilitate. They overlayed new surfaces and built new structures. Hence it is possible there was a time without permanent occupation between the two phases [40:138], but we leave this to future research. This composite model is not sensitive to intra-site variability, as there are not yet enough dates to do this reliably. The model is a reliable reflection of site-wide trends and estimates a precise boundary between permanent and temporary occupation: *~AD 1010 (970–1060, 95%)*. This is the most

important result of this paper, since the lack of permanent residents is our clearest material indicator of political and social disintegration at the site. The Bayesian models allow us to estimate a 95% probability range of eighty years; the 68% probability range covers just forty, *~AD 990–1030*. This precise estimate could indicate 1) the timing of a single depopulation event or 2) the central point of a staggered abandonment that lasted multiple decades. We cannot confidently estimate a duration for the period when people were leaving their homes, but it is clear that the process was no more than a few decades.

After permanent residence ended, there are deposits that were most likely left by visitors. This phase includes six dated contexts. In addition to the three from Akapana East and one from sector D in Mollo Kontu, we included one from sector A in Mollo Kontu (AA-275875) and a context atop the Akapana. This final context is a large camelid offering *~AD 1030* (SMU-2473 & INAH-972). Since it overlies melted adobe walls, it post-dates residential occupation. This brief post-collapse phase lasted around one generation until *~AD 1050 (Ending boundary)*, but imprecise dates leave open the possibility of later camps. We include all of these dates in the model, even though the latest one has a low agreement index. There are similar contexts in other sectors, such as Muru Ut Pata, but most remain undated, since previous projects have not targeted this phase. Future research could focus on these contexts to better understand the pace of abandonment and intra-site variability.

## Monument building and community architecture projects

The earliest monuments were built during the Late Formative Period, shortly after the site's initial occupation in the late AD 100s, based on recalibrated dates [44:107–111, 67, 68]. Next, work on the site's most ambitious monuments, the Pumapunku and the Akapana, was done mostly in the AD 600s. The models for the construction of the Pumapunku and the Akapana have very similar spans, with dates concentrated in the AD 600s. Since the samples are small, the boundaries have wide probability distributions. Toward a site-wide perspective, we combined these 11 dated events into a single model for monument building, which has boundaries of *~AD 590–720*. At the Pumapunku, the initial plans were abandoned early on and construction continued using reused stone beginning around *~AD 630*. At this same time, the construction of the Akapana also began with reused stone. After a gap in dated events associated with community architecture, there are three dated events in the tenth century. Two dated events from the Putuni can be reliably associated with the monumental blocks surrounding the courtyard, which were mostly moved from older buildings. At around the same time, the area near the Akapana was resurfaced and conduits were built into the new surface. These three events have medians of *~AD 920–970*, shortly before permanent residents left the site. There are other undated surfaces and monuments [52:79]. Overall, the continual reuse of building materials makes it challenging to precisely track building traditions and community architecture.

## Making and discarding redwares

Tiwanaku's decorated ceramics are highly recognizable and found at many sites in the region. To track these ceramics at Tiwanaku, we grouped all dated contexts with associated redwares in a single-phase model. This model suggests redwares were first discarded around *~AD 620 (570–650, 95%)*, and were probably being made for the first time alongside the first residential occupations, tombs, and the first use of reused blocks at the Pumapunku and Akapana. Just prior to this, there was a rush of migration to Tiwanaku [79], so the most likely scenario is that families first migrated to Tiwanaku and only then innovated this new style of pottery, which became an iconic part of Tiwanaku identity. The model places the last deposit of redwares

*~AD 1040 (1010–1080, 95%)*. The latest contexts with discarded redwares are from temporary occupations, a few decades after permanent residence ended. It is possible that people used earlier ceramics as heirlooms, or unintentionally mixed sherds from earlier contexts, for example, at the Pumapunku, earlier offerings were disturbed and possibly removed around *~AD 1020* (AA-68184) [76:140]. These late secondary uses of redwares are the final chapter of centuries-old networks that moved these vessels and their shared meanings around the Andes. At Tiwanaku, these ceramics were in circulation for *~410 years (370–450, 95%)*, a long period during which they became iconic material markers of enduring local and regional communities.

## Placing bodies

Placing human bodies is the most tenacious tradition at Tiwanaku. Enduring communities turned the area around the Akapana into a place reminiscent of a community burial ground. This tradition has its own chronology that does not directly track political integration. We built three single-phase models of very different contexts: tombs, violent deaths, and all other contexts with human bone, which have no burial architecture or signs of violence, and usually, no grave goods. The use of formal tombs ended just as violent deaths began *~AD 910*, a major watershed in the site's history that has been unrecognized until now. Violent deaths end *~AD 1020*, very close in time to the end of permanent residence and the use of redwares. Less formal burials are more common, both before and after this inflection point of social collapse; there are 22 dated contexts. In many cases, it seems likely that these bodies were left without much fanfare or by visitors, especially in the period after any permanent residents lived at the site. Many such contexts may not be directly relevant for understanding past political and social networks; we suggest that future sampling efforts be wary of less informative contexts.

**Tombs.** There are 12 dated individuals from tombs (Fig 4A–4C) and notably, all are near domestic structures. Like in many cultures in the world, the living and the dead shared domestic space. This practice also implies a degree of continuity, as the living family members could care for these people and invested more time in making the spaces and placing grave goods. In Akapana East, bodies were defleshed, probably as part of ancestor worship, but these are undated [32:442]. Ancestors and their homes, even after death, would have been key nodes in the networks that built multi-generational communities at Tiwanaku. This practice spanned *~AD 600–910*, according to a single-phase Bayesian model. Building tombs was a long-standing practice that ended three to four generations prior to the end of permanent residence and using redwares. It remains unknown how or where people were buried after this point. When tombs are no longer used, we reach a major inflection point in the site's history, since violent deaths begin. Boundaries of these two independent models have the same median: *~AD 910*, though error ranges are large.

**Violent deaths.** These burials follow the material patterns of ritualized murder [80]: they were found face down or bound with evidence of cutting, blunt trauma, or wounds to the head (Fig 4D–4I) [32, 56]. They were left in refuse pits without burial architecture or grave goods. One of these individual's DNA suggests she was from the Amazon, perhaps a captive since her hands were bound behind her back (TW056). Two dates are associated with two large group burials: seven people in Mollo Kontu and 16 individuals northwest of the Akapana. It is notable that this activity is not present elsewhere in the site nor at any other time. We have dates from seven individuals and associated contexts, which are in turn associated with many more individuals that suffered violent, or at the least, disrespectful deaths. Modeled as a single phase, the modeled medians are mostly in the mid AD 900s, with an ending boundary of *~AD 1020 (980–1090, 95%)*. This ending boundary strongly overlaps with the end of permanent

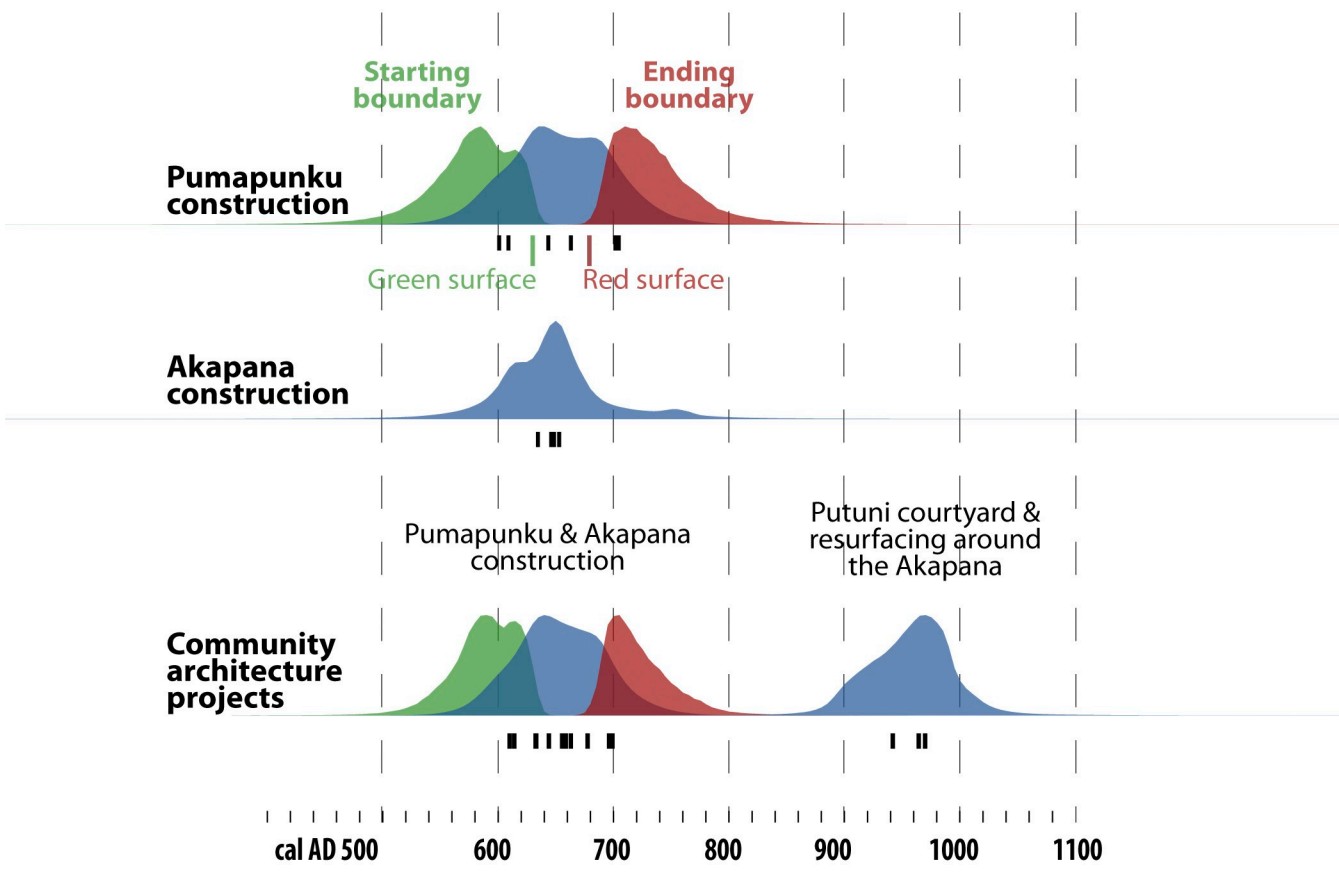

**Fig 7. Composite model for monuments and community construction projects.** (As in Fig 6).

residence and since both are fairly precise, it is likely they happened within a few decades. This sample is small but there is a clear temporal trend: these unusual deaths all happened during the final century of permanent residence (Fig 8). This surprising and novel result may be a crucial clue to understanding unraveling social networks.

## Discussion and summary

These results help us refine a narrative of changing practices that have been lumped together as 'collapse'. First, monument building at the Pumapunku and Akapana took place well before collapse, similar to other places with multiple large monuments: Egypt, Teotihuacan, and Rome [81–83]. Tiwanaku's early monuments became quarries for smaller projects as history began accelerating in the AD 900s.

The first major loss of existing traditions was when people stopped placing burials in formal tombs ~*AD 910*. This tradition lasted for around eleven generations, marking the end of a culturally-significant tradition. The community no longer saw the site as an acceptable place to leave ancestors to be visited and honored, nor use former residences as resting places for the dead. Violent burials began also began at ~*AD 910*, which were short-lived and unprecedented at the site. One interpretation is that elites were making a show of power by making public sacrifices and leaving them exposed to the elements [23:127–128, 32:443–446]. However, data here suggests that such events only took place during the final generations of residential occupation. They could have been final pleas to the gods in the face of crisis. Some victims may

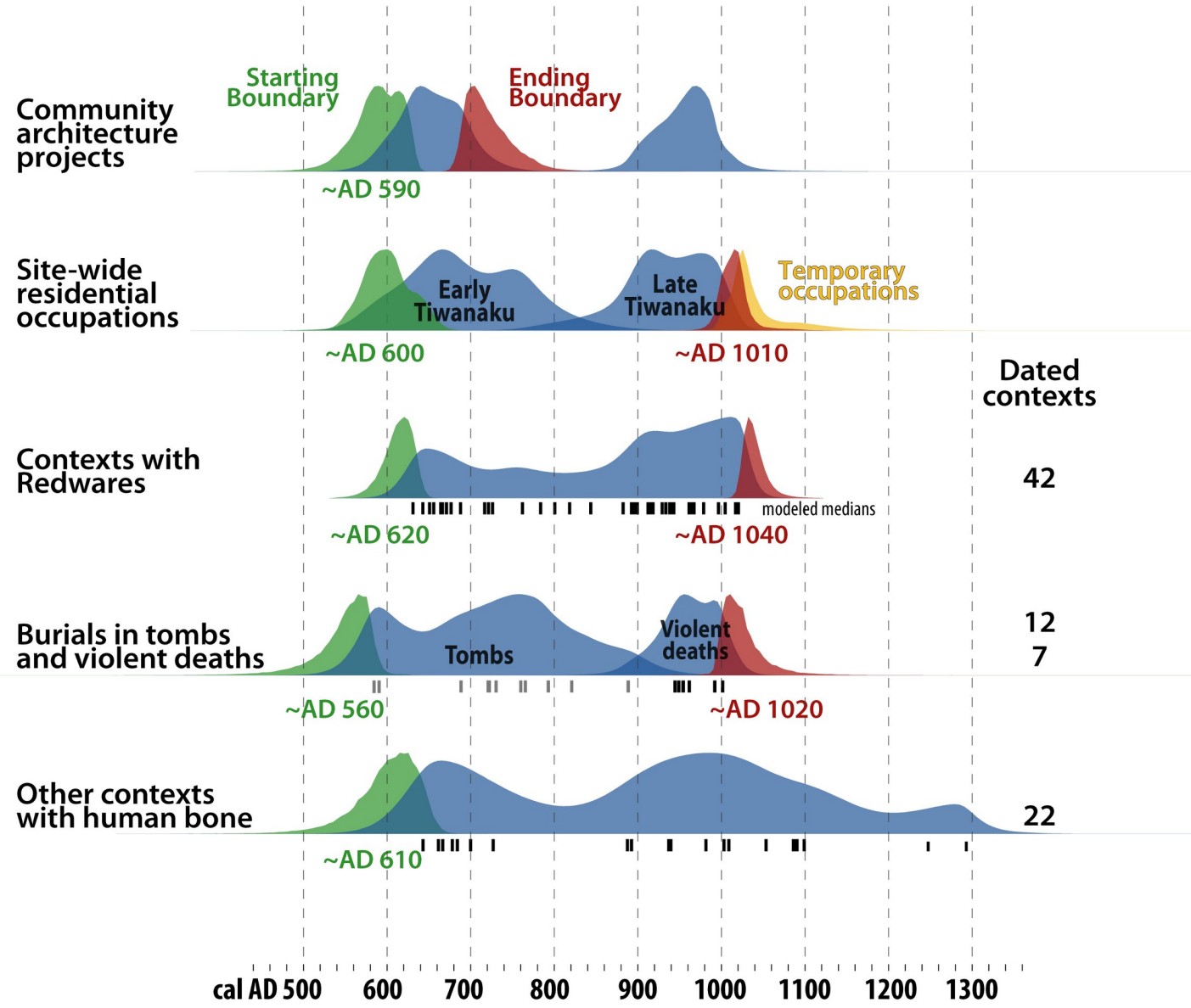

**Fig 8. Bayesian models of community architecture projects and site-wide residential occupations (Figs 6 and 7).** This Fig shows four single-phase models for contexts with redwares, burials in tombs, violent deaths (these are shown together but the models are independent), and all other contexts with human bone. The timing of collapse in the early eleventh century is best documented by the near-synchronous end to permanent residence, violent deaths, and the use and reuse of redwares (red probability curves). As in Fig 6.

have been captives from raids, but there is no indication of elites organizing violence on a larger scale. There are no indications of a strongly hierarchical society, instead there is a notable absence of redistributive features such as storage buildings, princely burials, formal roads, and way stations [84:155–157, 85]. There are no walls or defensive architecture, here or at any Tiwanaku site. There was no change in military technology, there are no barracks or concentrations of weapons, and projectile points are present but in low quantities [86:203]. There are images of trophy heads, but this potent image contrasts with the conspicuous absence of any real trophy heads [87]. Violence was present, but only for a short period near the end of residential occupation.

The end of violent burials ~*AD 1020* coincides with the end of some 17 generations of permanent residence, not counting a possible hiatus, around ~*AD 1010*. The ending boundaries strongly overlap and were probably contemporaneous. The site's permanent residents during the AD 900s are likely candidates for organizing raids and violent deaths. These residents probably coordinated the building of the Putuni compound and resurfacing around the Akapana. They pillaged the old to build the new. The city consumed itself, calling to mind the ouroboros serpent eating its own tail. A new, less stable political ideology may have emerged a few generations before collapse, which is comparable to other global examples such as Teotihuacan and Egypt.

The most important inflection point for major and rapid social change was around ~*AD 1010*, when violent burials and permanent residence ended. Residential areas were found with carbonized grass roofs, suggesting they were burned down. Some monoliths were vandalized or beheaded, but the timing is not clear [44, 45]. Metalworking was a part of life at Tiwanaku, and at least on a regional scale, this ended roughly around AD 950 [88]. The coeval end of these diverse practices is a robust indicator of rapidly unraveling social networks that ended around ~*AD 1010*, when people stopped living at the site.

Next, temporary camps last until ~*AD 1050*, though later camps are possible. These contexts include redwares, which were used at the site until ~*AD 1040*. The end of temporary camps and the use of redwares is about one generation after the end of permanent residence. Finally, the practice of leaving burials at Tiwanaku with no tombs or grave goods covers many centuries, from before and after residential collapse. Nearly all of these are prior to AD 1050, and many of these were probably placed by visitors. A few much later burials suggest the site continued to hold meaning long after its rapid collapse.

To return to the questions at the beginning of this paper, we can begin to address "what changed quickly?" We have tracked the end of permanent residence, temporary residence, using and discarding redwares, and violent deaths. To track the pace of change, we grouped these four ending boundaries as a phase, which has boundaries of ~*AD 1010–1050*, bookends to a process that likely lasted just *~20 years (0–70 years, 95%)*. This precise result suggests that these changes took place within a few decades, consistent with expectations for collapse [14]. This human-scale chronology supports Owen's [29] suggestion that collapse was explosive, at least at the city of Tiwanaku. We can speculate that the people involved were aware of the cascading changes they and their families were driving, intentionally or not. Nearly all of them decided to stop making and using redwares, and most prominently, to no longer reside here, or even visit– a profound and sudden shift at one of the Andes' most meaningful places.

## Regional comparisons and conclusion

Regional networks were left without their most potent node, the city of Tiwanaku, but this did not mean the collapse of all networks. Tiwanaku's chronology (Table 1) does not always align with dates from other sites, which raises intriguing questions for future research. For example,

**Table 1. Major inflection points in Tiwanaku's history.**

| Event | median (AD) | 95% |
|---|---|---|
| Site founded | 180 | 40–320 |
| Many new residences built and occupied by people with redwares | 600 | 540–670 |
| Last major monument construction | 720 | 670–790 |
| Last permanent residents | 1010 | 970–1050 |
| First Inca deposits in the Pumapunku | 1510 | 1440–1630 |

just as tombs are no longer placed at Tiwanaku, there is a surge of residential occupation and tombs in Moquegua near the Pacific coast [89]. These occupations end at the same time as at Tiwanaku, implying people living at both places were part of the same networks. Almost immediately, new post-collapse material traditions emerged specific to Moquegua [29, 89, 90], which did not happen around Tiwanaku. At Tiwanaku, redwares are no longer used after ~*AD 1040*, and this probably applies to most sites in the region. However, there is a key exception that speaks to the resilience of some traditions. At Lukurmata and Tiraska, closer to Lake Titicaca, redwares were used for several more generations [91:Table 3.1, 92, 93]. The nearby raised fields were also used well after Tiwanaku had no permanent residents, suggesting that a small community carried on Tiwanaku's ancient farming and potting traditions. Importantly, we cannot treat this as a continuation of a unified polity and argue that a state continued to operate into the twelfth century nor that polity-wide collapse was slow [cf. 23, 28, 93]. Instead, we suggest that after a resounding and explosive end to practices and residential life at Tiwanaku, a small resilient community in the Katari Valley continued some traditions on a smaller scale. We suggest future research build narratives that focus on these data patterns, and less so on generalized expectations for archaic states.

The specter of drought continues to loom over Tiwanaku's collapse, but this chronology suggests there was no direct relationship. We document a rapid social collapse many generations before the twelfth century, when most studies argue for reduced precipitation [94–97]. Climate reconstructions are still being refined and the most recent lake-level reconstruction shows a relatively minor decline prior to AD 1000, not a major drought. This study concludes that "although reduced precipitation . . . likely affected Tiwanaku's residents, it does not imply a drought-induced collapse" [98:7]. Future research should continue to consider the impact of environmental changes; for example, this minor decline in lake level may coincide with other proxies for aridity and dust [27, 88, 99–102]. For some age models, temporal differences are minor and error ranges overlap for both archaeological and climatic chronologies. Climate chronologies tend to be less precise than archaeological ones, a crucial but often-overlooked issue in any formal comparison [21, 22, 34]. Clearer sequences of events that duly consider error ranges are key for building human-scale sequences of cause and effect, especially in order to understand human responses to environmental changes.

Tiwanaku was one of the Andes' first cities and in the local Aymara language, its name means "the stone in the middle" [103:100]. For centuries, this lithic epicenter anchored communities who came together to live in residential complexes, build monuments, and leave the dead. Bayesian models show that in the early AD 1000s, these social networks unraveled quickly, probably within a generation. The stone in the center could no longer hold.

## Supporting information

**S1 Table. Tiwanaku radiocarbon dates.**
(XLSX)

**S1 File. Materials and methods details.**
(PDF)

**S2 File. Description of depositional sequences.**
(PDF)

**S3 File. OxCal code.**
(PDF)

**S4 File. OxCal files (code, outputs, and exported posteriors; see S1 and S3 Files).**
(ZIP)

**S5 File. Orthomosaic of 1930s aerial photographs.** Available from https://osf.io/v6j7n/.
(PDF)

**S6 File. Inclusivity questionnaire.**
(DOCX)

## Acknowledgments

We are grateful to colleagues who provided unpublished data, difficult-to-find references, and nurtured our thinking over the years: John Janusek (†), Linda Manzanilla, Giles Morrow, Andy Roddick, Scott Smith, and Jason Yaeger. Scans of Bennett's aerial photographs were provided courtesy of 1) Anna Guengerich, 2) Erwin Wodarczak and Candice Bjur, University Archives, University of British Columbia Library, Vancouver and 3) Maureen White, Division of Anthropology, Peabody Museum of Natural History, Yale University (see S5 File). Thanks to Andrew Millard and Christopher Bronk Ramsey for help with the OxCal inline arrays. The authors would like to express their gratitude to the *Centro de Investigaciones Arqueológicas, Antropológicas y Administración de Tiwanaku* (CIAAAT) for providing access to the facilities and research collections at Tiwanaku. We are also grateful to state and local organizations in Bolivia for granting permission to carry out archaeological research at Tiwanaku, including CIAAAT, the *Unidad de Arqueología y Museos*, *Ministerio de Cultura*, the community of Wankollo, the municipality of Tiahuanaco, and the *Consejo de Ayllus y Comunidades Originarios de Tiwanaku*. Finally, we are profoundly grateful to our friends in the town of Tiwanaku and the many Bolivian and North American students and colleagues who have worked with us.

## Author Contributions

**Conceptualization:** Erik J. Marsh, Alexei Vranich.

**Data curation:** Erik J. Marsh, Kelly J. Knudson.

**Formal analysis:** Erik J. Marsh, Santiago Ancapichún, Gianni Cunietti.

**Funding acquisition:** Alexei Vranich, Deborah Blom, Nicole C. Couture, Kelly J. Knudson, Danijela Popović.

**Investigation:** Alexei Vranich, Deborah Blom, Maria Bruno, Katharine Davis, Jonah Augustine, Nicole C. Couture.

**Methodology:** Erik J. Marsh.

**Project administration:** Alexei Vranich, Deborah Blom, Maria Bruno.

**Software:** Erik J. Marsh.

**Visualization:** Erik J. Marsh, Jonah Augustine, Santiago Ancapichún, Gianni Cunietti.

**Writing – original draft:** Erik J. Marsh, Alexei Vranich, Katharine Davis, Jonah Augustine.

**Writing – review & editing:** Erik J. Marsh, Alexei Vranich, Deborah Blom, Maria Bruno, Katharine Davis, Jonah Augustine, Nicole C. Couture, Santiago Ancapichún, Kelly J. Knudson, Danijela Popović, Gianni Cunietti.

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
