## [Decision Letter · Decision Letter 0]

15 May 2023

PONE-D-23-08915The center cannot hold: A Bayesian chronology for the collapse of TiwanakuPLOS ONE

Dear Dr. Marsh,

Thank you for submitting your manuscript to PLOS ONE. After careful consideration, we feel that it has merit but does not fully meet PLOS ONE’s publication criteria as it currently stands. Therefore, we invite you to submit a revised version of the manuscript that addresses the points raised during the review process.

We look forward to receiving your revised manuscript.

Kind regards,

John P. Hart, Ph.D.

Academic Editor

PLOS ONE

Journal Requirements:

5. We note that Figures 1 and 2 in your submission contain map/satellite images which may be copyrighted. All PLOS content is published under the Creative Commons Attribution License (CC BY 4.0), which means that the manuscript, images, and Supporting Information files will be freely available online, and any third party is permitted to access, download, copy, distribute, and use these materials in any way, even commercially, with proper attribution. For these reasons, we cannot publish previously copyrighted maps or satellite images created using proprietary data, such as Google software (Google Maps, Street View, and Earth). For more information, see our copyright guidelines: http://journals.plos.org/plosone/s/licenses-and-copyright.

a. You may seek permission from the original copyright holder of Figures 1 and 2 to publish the content specifically under the CC BY 4.0 license.  

Additional Editor Comments:

Both reviewers are very positive about your work. Reviewer 1 has several suggestions for improving the manuscript. Please take these into account while making your revisions.

Reviewers' comments:

Reviewer's Responses to Questions

**Comments to the Author**

1. Is the manuscript technically sound, and do the data support the conclusions?

Reviewer #1: Yes

Reviewer #2: Yes

2. Has the statistical analysis been performed appropriately and rigorously? 

Reviewer #1: Yes

Reviewer #2: Yes

3. Have the authors made all data underlying the findings in their manuscript fully available?

Reviewer #1: Yes

Reviewer #2: Yes

4. Is the manuscript presented in an intelligible fashion and written in standard English?

Reviewer #1: Yes

Reviewer #2: Yes

5. Review Comments to the Author

Reviewer #1: I welcome the opportunity to read and comment upon this important manuscript. In assembling and undertaking Bayesian analyses on a large sample of radiocarbon dates, the authors generate critical data and layout an intriguing and provocative narrative for the end of the site of Tiwanaku. The sample of 102 dates (45 of which are previously unpublished) from a range of contexts (and research projects) at the site are leveraged effectively to provide what the authors describe as a “generational scale chronology.” The Bayesian modeling adds further weight to existing arguments (including by the authors) that major changes occurred at the site of Tiwanaku before reduced regional precipitation. The authors make a strong case for this claim in this manuscript; however, the article’s impact will be significant beyond its (important) contribution to longstanding disagreements about the role of climate stress in Tiwanaku’s ‘collapse.’

This is a manuscript that should be published and my review of it is overwhelmingly positive. In the following, I make comments on specific sections. These are intended as suggestions to further strengthen an already impressive piece of work.

Abstract:

This needs to be rewritten. Although it describes the manuscript’s goals, data, and findings the prose is messy and confusing and confusing in places.

Introduction

I understand why the authors set up the manuscript in the way they do, given PLOSONE’s broad readership, one that isn’t limited to archaeologists, Tiwanaku scholars, much less scholars of Tiwanaku ‘collapse.’ Nonetheless, much of the critique both of narratives of collapse and of scholarly treatments of Tiwanaku has been made many times over in many venues, and the literature in both has really moved on from the way each is presented here. Consequently, while the implication that collapse is still characterized as all encompassing and that Tiwanaku is understood as “the capital of a politically-unified state” are helpful strawmen for setting the stage for the rest of the manuscript, they are not really reflective of current scholarship on either.

The authors are intentional about laying out material proxies for Tiwanaku’s ‘collapse’, namely by focusing on archaeologically visible discontinuities in long term community practices: 1) residential occupation, 2) monumental construction, 3) redwares, 4) treatment of human remains. This intent is well taken as is the concern that current comparative definitions of collapse as a reduction in political complexity are challenging. However, I am unpersuaded that what is proposed as being more explicit about collapse is really what the comparative scholarship means by collapse. The authors emphasize rapidity, but collapse (as it is comparatively understood) isn’t necessarily rapid. They emphasize the end of permanent residence and rapid abandonment, but the wider scholarship doesn’t support that as (comparatively) a marker of ‘collapse.’ More specifically, I question using the end of permanent residence as a proxy for social networks breaking down – I might be missing the connection being made here, but this could do with clarification.

I would suggest adding to the introduction the very clear and important point that these data and interpretations focus specifically on the site of Tiwanaku. Although mentioned in the introduction, and elaborated upon at the end of the manuscript in an important section on continuity/resilience at other sites both in the Tiwanaku heartland and in the Moquegua, this needs to be clearly articulated in the introduction. This is particularly important given that production and consumption of redwares are one of the long-term community practices that are tracked as a proxy for collapse. Elsewhere, the manufacture and use of redwares that are sometimes indistinguishable from pre-‘collapse’ Tiwanaku ceramics continues in the face of other significant societal transformation. The authors do make clear in their final section that their chronology and narrative is specific to the site of Tiwanaku but I’d encourage including this in the introduction.

Materials & Methods

This is a very robust data set. I am not a radiocarbon specialist so I defer to others on the use of a custom calibration curve, but to a non-specialist it is clearly explained and justified.

Query: how large is the set of unpublished dates mentioned in line 226?

Results

Results are largely laid out clearly and easy to follow. A few minor suggestions/corrections.

The sections on Pumapunku and the Akapana cross reference dates mentioned in the text with the sample number in parentheses (e.g. TIW002) but this isn’t done for other contexts. It would be helpful to add that in for all individually mentioned dates.

Line 297: change “recalls” to “recall”

Line 306: change “distributed” to “disturbed”

Line 324-325: as discussed, I have some concerns about the use of redwares as a marker for end/collapse. There does seem to be a case for this at Tiwanaku but as already mentioned (and as discussed by the authors at the end of the manuscript), Tiwanaku style redwares continued to be made and used in many other Tiwanaku derived contexts after other major social transformations. Again, I think that really reiterating in the introduction that this is a narrative about the site of Tiwanaku should be done.

An additional concern about redwares is that of course ceramic vessels can be heirlooms – is there any danger of dating contexts too late? Just a thought.

Related, under the discussion of Akapana contexts, one is a context with undecorated ceramics – is this taken to be pre or post collapse?

Line 411: change ( to [

Line 432: “blue and small projectile points” – is this sentence missing something, it was unclear to me what was meant.

Line 443: “permanent residences is our most reliable proxy for collapse”. Again, I question this a little given that although urban residence can shift or rework during processes of ‘collapse’, comparative data does not support the claim that permanent residence ends at major settlements. Given that PLOSONE has a broad readership and there is potential that non-specialist readers will take this as a universal archaeological correlate of ‘collapse.’ I have no concern with it being used for Tiwanaku but I think again clarifying that this is a correlate for the site of Tiwanaku is important.

The most striking thing to this reviewer in this manuscript is the generational chronology around violent death. As the authors discuss this inflection point (both the beginning and the end) has not been identified before and I would suggest is the most significant contribution of the manuscript, meriting greater attention,

Discussion

Lines 611-612: this reads as another slightly dated strawman given that current narratives of Tiwanaku do push back on expectations of archaic sites.

Supplementary Files

S2: pp 6, the line about redwares being distinguishable from post-Tiwanaku ceramics should again be clarified to be very clear that they are at Tiwanaku, as they are not in other Tiwanaku influenced sites and regions.

Figure S2.1 is cut off at the bottom

Reviewer #2: This article is very clear and well thought out. The goals for the analysis are four-fold (distinguish between short and long-term occupation, refine the chronologies of the Akapana and Pumapunku, address dated contexts with redware, and address changing practices related to the treatment of human remains.) The authors nicely tie together all these goals—along with the timing of the use and downfall of the residential areas—into a coherent argument. The authors nicely address the changing nature of practices and use of certain areas of Tiwanaku for the duration of its use. They also use Bayesian analysis to further refine the timing of the collapse. The multiple Bayesian models are clear in both intent and execution. The application of a mixed curve is appropriate for the site. The figures are also clear and contain the right amount of information in the captions. The supplemental materials provide code and further information on the Bayesian modeling. Although all the models are useful and necessary, the new findings around the timing of violent deaths vs. burials in tombs I find to be really compelling. Especially in light of understanding the somewhat rapid decline of the settlement. These data and models are also very welcome in finally putting to rest the theory of collapse due to climatic disturbances. We can now for certain say that the climatic unrest postdated whatever socio-political unrest that led to Tiwanaku’s demise. This article provides a very necessary summary of how we understand Tiwanaku chronology today and very solid foundation for future work at the site and beyond. I vote to publish as is. I don’t see any major flaws.

6. PLOS authors have the option to publish the peer review history of their article (what does this mean?). If published, this will include your full peer review and any attached files.

Reviewer #1: No

Reviewer #2: No

---

## [Author Response · Author response to Decision Letter 0]

28 Jun 2023

Response to reviewers

We provide the original comment followed by our response, beginning with a dash (–).

Editor: Both reviewers are very positive about your work. Reviewer 1 has several suggestions for improving the manuscript. Please take these into account while making your revisions.

– We are encouraged that both reviewers took a positive view of this paper. We thank them for their kind comments.

Reviewer #1: Abstract: This needs to be rewritten. Although it describes the manuscript’s goals, data, and findings the prose is messy and confusing and confusing in places. 

– We rewrote the abstract and consulted a professional editor.

 I understand why the authors set up the manuscript in the way they do, given PLOSONE’s broad readership, one that isn’t limited to archaeologists, Tiwanaku scholars, much less scholars of Tiwanaku ‘collapse.’ Nonetheless, much of the critique both of narratives of collapse and of scholarly treatments of Tiwanaku has been made many times over in many venues, and the literature in both has really moved on from the way each is presented here. Consequently, while the implication that collapse is still characterized as all encompassing and that Tiwanaku is understood as “the capital of a politically-unified state” are helpful strawmen for setting the stage for the rest of the manuscript, they are not really reflective of current scholarship on either. 

– The reviewer is correct, we mostly targeted non-specialist readers, even non-archaeologists, who have heard of some of these older theories that are not currently scholars of Tiwanaku's collapse. We attempted to summarize this very briefly. Moreover, we find this necessary because we do not think the non-archaeological literature has completely moved on (unfortunately). For example, in a recent paper in Quaternary Science Reviews (Arnold et al. 2022), climate scientists revive dated and simplistic arguments, essentially drought=collapse, suggesting to us that for non-archaeologists, these older "common sense" theories are quite relevant to argue against.

– We have made changes to the introduction to clarify that our data can only speak to the city of Tiwanaku, not a politically-unified state. This is essential, because we maintain that the end of permanent residence is a very clear indicator of a city's collapse, but take the reviewer's point that this is not may not reflect a polity's collapse. We feel this is the main clarification that needs to be made to address the reviewer's concerns.

– In line 102, we clarify: "We suggest it is important to take a site-by-site approach, so our argument for collapse in this paper applies to the site of Tiwanaku, not all sites with similar ceramics." This is repeated in the conclusion.

 However, I am unpersuaded that what is proposed as being more explicit about collapse is really what the comparative scholarship means by collapse. The authors emphasize rapidity, but collapse (as it is comparatively understood) isn’t necessarily rapid. They emphasize the end of permanent residence and rapid abandonment, but the wider scholarship doesn’t support that as (comparatively) a marker of ‘collapse.’ More specifically, I question using the end of permanent residence as a proxy for social networks breaking down – I might be missing the connection being made here, but this could do with clarification. 

– It seems the reviewer is thinking of the collapse of a polity, not a city. We clarified this.

 I would suggest adding to the introduction the very clear and important point that these data and interpretations focus specifically on the site of Tiwanaku. Although mentioned in the introduction, and elaborated upon at the end of the manuscript in an important section on continuity/resilience at other sites both in the Tiwanaku heartland and in the Moquegua, this needs to be clearly articulated in the introduction. This is particularly important given that production and consumption of redwares are one of the long-term community practices that are tracked as a proxy for collapse. Elsewhere, the manufacture and use of redwares that are sometimes indistinguishable from pre-‘collapse’ Tiwanaku ceramics continues in the face of other significant societal transformation. The authors do make clear in their final section that their chronology and narrative is specific to the site of Tiwanaku but I’d encourage including this in the introduction. 

– This is a very helpful suggestion and we have made these changes to the introduction.

 Query: how large is the set of unpublished dates mentioned in line 226? 

– Thirteen. We refer the reviewer to the S1 File here. Since there is overlap between the counts of dates made by each project and for each sector, we did not specify this detail in the main text.

 

The sections on Pumapunku and the Akapana cross reference dates mentioned in the text with the sample number in parentheses (e.g. TIW002) but this isn’t done for other contexts. It would be helpful to add that in for all individually mentioned dates. 

– In the main text, we only refer to individuals, not dates. We clarified this. Most of these are in fact dated by two dates, which we specify in S1 and S3. We have moved almost all references of individual dates to S3, to make the text more readable.

 Line 297: change “recalls” to “recall” Line 306: change “distributed” to “disturbed"

Line 411: change ( to [

 Figure S2.1 is cut off at the bottom

–We fixed these errors.

 Line 324-325: as discussed, I have some concerns about the use of redwares as a marker for end/collapse. There does seem to be a case for this at Tiwanaku but as already mentioned (and as discussed by the authors at the end of the manuscript), Tiwanaku style redwares continued to be made and used in many other Tiwanaku derived contexts after other major social transformations. Again, I think that really reiterating in the introduction that this is a narrative about the site of Tiwanaku should be done. 

– See above.

 An additional concern about redwares is that of course ceramic vessels can be heirlooms – is there any danger of dating contexts too late? Just a thought. 

– We added this intriguing suggestion to line 481. If this is the case, it would have a very minor effect on the Bayesian models. This practice might affect formal tombs more, which were discontinued long before the site was abandoned.

 Related, under the discussion of Akapana contexts, one is a context with undecorated ceramics – is this taken to be pre or post collapse? 

– These are from the same structure mentioned in the previous sentence. We clarified this in line 358.

 Line 432: “blue and small projectile points” – is this sentence missing something, it was unclear to me what was meant. 

– Text was unintentionally deleted here. We have reviewed the source of the information and corrected the text (line 398).

 

Line 443: “permanent residences is our most reliable proxy for collapse”. Again, I question this a little given that although urban residence can shift or rework during processes of ‘collapse’, comparative data does not support the claim that permanent residence ends at major settlements. Given that PLOSONE has a broad readership and there is potential that non-specialist readers will take this as a universal archaeological correlate of ‘collapse.’ I have no concern with it being used for Tiwanaku but I think again clarifying that this is a correlate for the site of Tiwanaku is important. 

– This is correct – we only mean this as an indicator of collapse for the city of Tiwanaku. This is not meant to imply collapse at any other site. In the introduction, we have clarified what we mean by collapse – only of the city of Tiwanaku, not at multiple settlements. 

 The most striking thing to this reviewer in this manuscript is the generational chronology around violent death. As the authors discuss this inflection point (both the beginning and the end) has not been identified before and I would suggest is the most significant contribution of the manuscript, meriting greater attention.

– We have made additional comments about this finding, which was a surprise to us. We added it to the abstact.

 

Lines 611-612: this reads as another slightly dated strawman given that current narratives of Tiwanaku do push back on expectations of archaic sites. 

– We modified this sentence with a more constructive attitude and moved it to the conclusion. It now reads "We suggest future research build narratives that focus on these data patterns, and less so on general expectations for archaic states."

 Supplementary Files S2: pp 6, the line about redwares being distinguishable from post-Tiwanaku ceramics should again be clarified to be very clear that they are at Tiwanaku, as they are not in other Tiwanaku influenced sites and regions. 

– We made this clarification.

---

## [Editor Report · Decision Letter 1]

5 Jul 2023

The center cannot hold: A Bayesian chronology for the collapse of Tiwanaku

PONE-D-23-08915R1

Dear Dr. Marsh,

We’re pleased to inform you that your manuscript has been judged scientifically suitable for publication and will be formally accepted for publication once it meets all outstanding technical requirements.

Kind regards,

John P. Hart, Ph.D.

Academic Editor

PLOS ONE
---

## [Editor Report · Acceptance letter]

17 Aug 2023

PONE-D-23-08915R1 

The center cannot hold: A Bayesian chronology for the collapse of Tiwanaku 

Dear Dr. Marsh:

I'm pleased to inform you that your manuscript has been deemed suitable for publication in PLOS ONE. Congratulations! Your manuscript is now with our production department. 

Kind regards, 

on behalf of

Dr. John P. Hart 

Academic Editor

PLOS ONE